# BALANCED CLUSTERING IN REDUCED DIMENSIONS

## ABSTRACT

High-dimensional data often require embedding into lower-dimensional spaces to preserve essential features and structures, which is critical for large-scale analysis. Existing approaches typically treat embedding and clustering as joint optimization tasks but fail to integrate them within a unified framework, limiting clustering performance. Moreover, the interplay between labels and manifold structure is frequently overlooked. To address these challenges, we propose a low-dimensional manifold clustering method that integrates K-means with manifold learning. To mitigate inaccuracies in initial cluster labels, we introduce neighborhood constraints that promote intra-class compactness and inter-class separation, thereby improving label reliability. These refined labels are then used to construct a manifold representation, which in turn enhances clustering in a self-supervised loop that enforces consistency between structure and labels. Notably, we show that maximizing the Schatten-p norm naturally preserves class balance, and we provide a theoretical justification for this property. Extensive experiments on multiple datasets demonstrate the effectiveness and robustness of our approach.

## 1 INTRODUCTION

The objective of clustering is to partition data into multiple clusters, ensuring that samples within the same cluster exhibit high degrees of similarity while those in different clusters differ significantly. However, in high-dimensional spaces, clustering faces the "curse of dimensionality." Due to the sparse distances between samples, traditional algorithms struggle to identify similarities. Additionally, noise and redundant information in high-dimensional data complicate the clustering process, further hindering the accuracy of the resultsXie et al. (2018); Guardieiro et al. (2025); Atzberger et al. (2025). Therefore, dimensionality reduction clustering algorithms have emergedXu et al. (2017); Jeon et al. (2024).

A common approach to dimensionality reduction clustering is to first reduce dimensionality and then apply clustering (e.g., PCAKM). However, separating these steps can result in features unsuitable for clustering. To better preserve geometric structure and discriminative information, various subspace clustering methods have been proposed Elhamifar & Vidal (2009); Liang et al. (2019); Li et al. (2017); Fu et al. (2021); Zhang et al. (2020a); Yang et al. (2020); Elhamifar & Vidal (2013); Wang et al. (2022). For example, Elhamifar et al.Elhamifar & Vidal (2009; 2013) proposed sparse subspace clustering by building similarity graphs from sparse coefficients, capturing local but not global structures. Wang et al.Wang et al. (2018b) improved this by adapting the affinity matrix during subspace learning. Fu et al.Fu et al. (2021) further incorporated projective distance and Laplacian constraints to better model both global and local structures. Liu et al.Liu et al. (2023) tackled feature redundancy by projecting data before subspace representation. However, the segregation of dimensionality reduction and clustering may lead to extracted features that are unsuitable for the clustering task, resulting in subpar clustering performance.

Some one-step dimensionality reduction clustering methods have been developed to address these challenges. Ding et al. Ding & Li (2007) alternated LDA and K-means processes, performing K-means in the embedding space to obtain labels to guide LDA optimization. However, this alternating strategy may cause non-convergence problems. Wang et al. Wang et al. (2021) and Wang et al. Wang et al. (2023) proved the equivalence between LDA and K-means, rewriting LDA as an unsupervised LDA model to achieve clustering and low-dimensional embedding synchronization. Additionally, Nie et al. Nie et al. (2023) revealed the equivalence of discriminative dimensionality reduction clustering and LDA using the regression form of LDA Nie et al. (2012). Hou et al. Hou et al. (2015)

proposed a unified one-step dimensionality reduction clustering framework, which preserves the main statistical information of the data through PCA and applies K-means clustering in the subspace, optimizing them alternately to ensure reliable clustering results while reducing the complexity of the data. Wang et al. Wang et al. (2019) unified LPP and K-means to achieve dimensionality reduction clustering based on local structure through a balanced hyperparameter. However, when dealing with more complex cluster structures, it may be limited by local linear relationships. To solve this problem, Zhou et al. Zhou et al. (2021) generated an orthogonal projection space to maintain the local structure of the data and realized clustering in a low-dimensional space.

Although previous one-step methods have made progress, they only jointly optimize K-means and graph embedding, treating the manifold structure and clustering labels as independent. To address these limitations, we propose a self-supervised manifold clustering framework that integrates K-means and low-dimensional graph embedding (LPP) into a unified model. In this framework, cluster labels produced by K-means are used to construct the data manifold, which in turn guides the dimensionality reduction. To address potential inaccuracies in the initial labels, we introduce a neighborhood constraint: two samples are considered similar only if they share the same label and are mutual nearest neighbors. This constraint improves label reliability and ensures consistency between the manifold structure and clustering assignments. Moreover, unlike the commonly used the Schatten-p normWang et al. (2018a) for matrix completionYang et al. (2022), image restorationXie et al. (2016); Zhang et al. (2020b), we introduce the maximization of the Schatten-p norm to ensure cluster balance. Our contributions are summarized as follows:

(1) **Regarding Model Fusion:** The existing dimensionality reduction clustering algorithm is only to find the optimal combination of manifold learning and K-means, rather than structurally unifying them. In contrast, we fuse them into a unified model to construct a self-supervised low-dimensional manifold clustering framework.

(2) **Regarding the Consistency of Labels and Manifold Structure:** The existing methods do not make full use of the relationship between manifold structure and labels. In contrast, we use labels and neighborhood constraints to guide the learning of manifold structure, and cluster update labels on the manifold structure. This cyclic iterative optimization process ensures the consistency of manifold structure and labels.

(3) **Regarding Cluster Balance:** The existing methods achieve matrix completion and image restoration by minimizing Schatten-p norm. In contrast, we discovered and proved that maximizing the Schatten-p norm during the clustering process can achieve clustering balance.

## 2 RELATED WORK

### 2.1 MANIFOLD K-MEANS CLUSTERING

Denote the sample matrix as $\mathbf{X} = [\mathbf{x}^1, \mathbf{x}^2, \ldots, \mathbf{x}^d] \in \mathbb{R}^{N \times d}$, where $N$ is the number of samples and $d$ is the feature dimension. The label matrix is $\mathbf{G} = [\mathbf{g}_1, \mathbf{g}_2, \ldots, \mathbf{g}_N]^T \in \mathbb{R}^{N \times c}$. K-means aims to divide the samples into $c$ clusters by iteratively determining the center of mass and minimizing the distance between the samples and the class centroids.

$$\min_{\mathbf{G}, \mathbf{u}^j} \sum_{i,j} g_{ij} \left\| \mathbf{x}^i - \mathbf{u}^j \right\|_2^2 \quad \text{s.t. } \mathbf{G} \in \text{Ind} \tag{1}$$

where $\mathbf{U} = [\mathbf{u}^1, \mathbf{u}^2, \ldots, \mathbf{u}^c] \in \mathbb{R}^{c \times d}$ is the centroid matrix. The discrete nature of the labels implies that if $\mathbf{x}^i$ belongs to the $j$-th cluster, then $g_{ij} = 1$; otherwise, $g_{ij} = 0$.

To avoid the influence of the initial centroids of K-means, Gao et al.Gao et al. (2025) re-expressed K-means from the perspective of manifold learning and formulated it as Eq.(2).

$$\min_{\mathbf{G}} \sum_{i=1}^{N} \sum_{l=1}^{N} \left\| \mathbf{x}^i - \mathbf{x}^l \right\|_2^2 s_{il} \quad \text{s.t. } \mathbf{G} \in Ind \tag{2}$$

where $\mathbf{S} = \mathbf{Q}\mathbf{Q}^T$, and $\mathbf{Q} = \mathbf{G}\mathbf{P}^{-1/2}$, $\mathbf{P} = \text{diag}(\sum_{i=1}^{N} g_{i1}, \sum_{i=1}^{N} g_{i2}, \ldots, \sum_{i=1}^{N} g_{ic})$.

## 2.2 GRAPH EMBEDDING FRAMEWORK

For dimensionality reduction tasks, LDA Belhumeur et al. (1997a), PCA Turk & Pentland (1991) and LPP He & Niyogi (2003) have different motivations, but they all aim to effectively reduce data dimensions while maximizing the retention of core information and structural features in the original data. From this unified perspective, all of the above dimensionality reduction methods can be represented within a unified graph embedding framework Yan et al. (2005):

$$\min_{\mathbf{M}} \frac{\mathrm{tr}(\mathbf{M}^T\mathbf{X}^T\mathbf{L}\mathbf{X}\mathbf{M})}{\mathrm{tr}(\mathbf{M}^T\mathbf{X}^T\mathbf{B}\mathbf{X}\mathbf{M})} \Rightarrow \min_{\mathbf{M}} \sum_{i=1}^{n}\sum_{j=1}^{n} \left\| \mathbf{x}^i\mathbf{M} - \mathbf{x}^j\mathbf{M} \right\|_2^2 s_{ij} \quad \text{s.t. } \mathbf{M}^T\mathbf{X}^T\mathbf{B}\mathbf{X}\mathbf{W} = \mathbf{con} \tag{3}$$

where $\mathbf{M} \in \mathbb{R}^{d \times m}$ is the projection matrix, and $m$ is the dimension after dimensionality reduction. The symmetric matrix $\mathbf{S} \in \mathbb{R}^{N \times N}$ represents the similarity matrix. $\mathbf{L} = \widetilde{\mathbf{D}} - \mathbf{S}$ is the Laplacian matrix, where $\widetilde{\mathbf{D}}_{(ii)} = \sum_{i \neq j} S_{ij}, \forall i$. $\mathbf{c}$ is a constant, and $\mathbf{B}$ is a constraint matrix.

The difference between these algorithms lies in calculating the similarity matrix $\mathbf{S}$ of the graph and selecting the constraint matrix $\mathbf{B}$. For LDA, $s_{ij} = \frac{\delta_{k_i,k_j}}{n_{k_i}}$, and $\mathbf{B} = \mathbf{I} - \frac{1}{N}ee^T$, where $e = [1,1,\ldots,1]^T \in \mathbb{R}^{N \times 1}$. For PCA, $s_{ij} = \frac{1}{N}$ for $i \neq j$, and $\mathbf{B} = \mathbf{I}$. For LPP, $s_{ij} = e^{-\frac{\|\mathbf{x}_i - \mathbf{x}_j\|_2^2}{\sigma}}$, and $\mathbf{B} = \widetilde{\mathbf{D}} = diag(\sum_{j=1}^{n} s_{1j}, \ldots, \sum_{j=1}^{n} s_{Nj})$.

## 3 METHODOLOGY

Most previous dimensionality reduction clustering methods usually start with manifold learning and then K-means clustering, or jointly optimize manifold learning and K-means, but they only search for the optimal combination of manifold learning and clustering, and fail to truly unify. Moreover, these methods usually treat the manifold structure $\mathbf{S}$ and clustering label $\mathbf{G}$ as independent, ignoring the potential interactions and dependencies between them. We integrate graph embedding and K-means into a unified framework to achieve manifold clustering in the reduced space. Specifically, we use the labels obtained from K-means clustering to construct the manifold structure of the data, which in turn guides the dimensionality reduction process. Considering that the labels obtained by clustering may not be completely accurate, we introduced neighborhood constraints when applying LPP to the framework, so that samples of the same class are closer in the neighborhood, while samples of different classes are pulled away to improve the reliability of labels. Therefore, the manifold structure is determined by the labels and the neighborhood relation, as shown in Eq. (4).

$$\min_{\mathbf{M},\mathbf{G}} \sum_{i=1}^{N}\sum_{l=1}^{N} \left\| \mathbf{x}^i\mathbf{M} - \mathbf{x}^l\mathbf{M} \right\|_2^2 s_{il} \quad \text{s.t. } \mathbf{G} \in \text{Ind}, \mathbf{M}^T\mathbf{X}^T\widetilde{\mathbf{D}}\mathbf{X}\mathbf{M} = \mathbf{I} \tag{4}$$

where $\mathbf{M} \in \mathbb{R}^{d \times m}$ is the projection matrix, and $m$ is the dimension after dimensionality reduction. $\widetilde{\mathbf{D}}$ is a diagonal matrix with $\widetilde{d}_{ii} = \sum_{l=1}^{N} s_{il}$. $\mathbf{S} \in \mathbb{R}^{N \times N}$ refers to the manifold structure constrained by $\mathbf{Q}$ and neighborhood information, defined as:

$$s_{il} = \begin{cases} \langle \mathbf{q}_i, \mathbf{q}_l \rangle, & \text{if } \mathbf{x}^i \in \mathcal{N}_k(\mathbf{x}^l) \text{ or } \mathbf{x}^l \in \mathcal{N}_k(\mathbf{x}^i); \\ 0, & \text{otherwise.} \end{cases} \tag{5}$$

where $\mathbf{Q} = \mathbf{G}\mathbf{P}^{-1/2}$, $\mathbf{P}$ is a diagonal matrix whose diagonal elements are defined as $p_{jj} = \sum_{i=1}^{N} g_{ij}$.

### 3.1 OPTIMIZATION

The optimization of Eq. (4) focuses on the solution of projection matrix $\mathbf{M}$ and label matrix $\mathbf{G}$.

• **Update M with fixed G:** In this case, the sub-problem for $\mathbf{M}$ can be formulated as follows:

$$\min_{\mathbf{M}} \sum_{i=1}^{N}\sum_{l=1}^{N} \left\| \mathbf{x}^i\mathbf{M} - \mathbf{x}^l\mathbf{M} \right\|_2^2 s_{il} \quad \text{s.t.} \mathbf{M}^T\mathbf{X}^T\widetilde{\mathbf{D}}\mathbf{X}\mathbf{M} = \mathbf{I} \tag{6}$$

As derived in He & Niyogi (2003), Eq. (6) can be rewritten as:

$$\min_{\mathbf{M}} \text{tr}(\mathbf{M}^T \mathbf{X}^T \mathbf{L}_S \mathbf{X} \mathbf{M}) \quad \text{s.t.} \mathbf{M}^T \mathbf{X}^T \widetilde{\mathbf{D}} \mathbf{X} \mathbf{M} = \mathbf{I} \tag{7}$$

where $\widetilde{\mathbf{D}}$ is a diagonal matrix with $\widetilde{d}_{ii} = \sum_{l=1}^{N} s_{il}$, and $\mathbf{L}_S = \widetilde{\mathbf{D}} - \mathbf{S}$.

Further, Eq. (7) can be transformed to solve:

$$\min_{\mathbf{M}^T \mathbf{M} = \mathbf{I}} \text{tr}(\mathbf{M}^T \mathbf{X}^T (\mathbf{L}_S - \eta \widetilde{\mathbf{D}}) \mathbf{X} \mathbf{M}) \tag{8}$$

where $\eta$ is an adjustable parameter. The optimal value of $\mathbf{M}$ is then composed of the eigenvectors corresponding to the first $m$ smallest eigenvalues of $\mathbf{X}^T (\mathbf{L}_S - \eta \widetilde{\mathbf{D}}) \mathbf{X}$. Algorithm 1 presents the pseudo-code for solving the problem (6).

---

**Algorithm 1:** Solving Problem (6)

---

**Require:** Data matrix $\mathbf{X} \in \mathbb{R}^{N \times d}$; Cluster assignment matrix $\mathbf{G} \in \mathbb{R}^{N \times c}$; Number of neighbors $k$.
**Ensure:** Projection matrix $\mathbf{M} \in \mathbb{R}^{d \times m}$.
 1: **while** *not converge* **do**
 2:     Determine manifold structure $\mathbf{S}$ by Eq.(5);
 3:     Take the eigenvectors corresponding to the first $m$ smallest eigenvalues of $\mathbf{X}^T (\mathbf{L}_S - \eta \widetilde{\mathbf{D}}) \mathbf{X}$
      to form $\mathbf{M}$;
 4: **end while**
 5: **return**: The projection matrix $\mathbf{M}$.

---

• **Update G with fixed M:** The sub-problem for $\mathbf{G}$ can be formulated as follows:

$$\min_{\mathbf{G} \in Ind} \sum_{i=1}^{N} \sum_{l=1}^{N} \left\| \mathbf{x}^i \mathbf{M} - \mathbf{x}^l \mathbf{M} \right\|_2^2 s_{il} \tag{9}$$

When solving $\mathbf{G}$, it can be further expressed as

$$\min_{\mathbf{G} \in Ind} \sum_i \sum_l e_{il} \langle \mathbf{q}_i, \mathbf{q}_l \rangle = \min_{\mathbf{G} \in Ind} \text{tr}(\mathbf{G}^T \mathbf{E} \mathbf{G} \mathbf{P}^{-1}) \tag{10}$$

where $\mathbf{Q} = \mathbf{G} \mathbf{P}^{-1/2}$, $e_{il} = \begin{cases} \left\| \mathbf{x}^i \mathbf{M} - \mathbf{x}^l \mathbf{M} \right\|_2^2, & \text{if } \mathbf{x}^i \in \mathcal{N}_k(\mathbf{x}^j) \text{ or } \mathbf{x}^j \in \mathcal{N}_k(\mathbf{x}^i); \\ 0, & \text{otherwise.} \end{cases}$

The model (10) is difficult to solve, so to optimize it and ensure cluster balance, we introduce Theorem 1.

**Theorem 1** *Let the total number of samples satisfies $N = \sum_{j=1}^{c} n_j$ with $n_j \geq 0$. Eq. (11) with $1 \leqslant p < 2$ attains its maximum value when $n_j = \frac{N}{c}$ for all $j$. At this optimum, $\mathbf{G}$ represents a discrete and balanced cluster assignment.*

$$\max_{\mathbf{G} \geq \mathbf{0}, \mathbf{G}\mathbf{1}=\mathbf{1}} \|\mathbf{G}\|_{sp}^p \tag{11}$$

**Proof 1**

$$\|\mathbf{G}\|_{sp}^p = \sum_{j=1}^{c} \sigma_j^p(\mathbf{G}) = \sum_{j=1}^{c} (\tau_j(\mathbf{G}^T \mathbf{G}))^{\frac{p}{2}} = \sum_{j=1}^{c} b_j^{\frac{p}{2}} \tag{12}$$

*where $b_j = \tau_j(\mathbf{G}^T \mathbf{G})$.*

*Let $\mathbf{b} = [b_1, b_2, \ldots, b_c]^T \in \mathbb{R}^{c \times 1}$, $\lambda_1 = \lambda_2 = \ldots = \lambda_c = \frac{1}{c}$. $f(b_j) = b_j^{\frac{p}{2}}$ is a concave function with respect to $b_j$, $1 \leqslant p < 2$, then according to Jensen inequality, we have*

$$f\left(\sum_{j=1}^{c} \lambda_j b_j\right) \geq \sum_{j=1}^{c} \lambda_j f(b_j) = \frac{1}{c} \sum_{j=1}^{c} f(b_j) = \frac{1}{c} \|\mathbf{G}\|_{sp}^p \tag{13}$$

*Equality holds if and only if $b_1 = b_2 = \ldots = b_c$.*

*The left side of inequality (13) can be simplified to*

$$f(\frac{1}{c}\sum_{j=1}^{c}\tau_j(\mathbf{G}^T\mathbf{G})) = f(\frac{1}{c}\|\mathbf{G}\|_F^2) = (\frac{1}{c}\|\mathbf{G}\|_F^2)^{\frac{p}{2}} \quad (14)$$

*In this case, in order to find the maximum of $\frac{1}{c}\|\mathbf{G}\|_{sp}^p$, we can translate to finding the maximum of $(\frac{1}{c}\|\mathbf{G}\|_F^2)^{\frac{p}{2}}$ under the constraint $b_1 = b_2 = \ldots = b_c$, i.e.,*

$$\max_{g_{ij}}\|\mathbf{G}\|_F^2 = \max_{g_{ij}}\sum_{ij}g_{ij}^2 = \max_{g_{ij}}\sum_i\sum_j g_{ij}^2 \quad s.t. g_{ij} \geq 0, \sum_j g_{ij} = \mathbf{1}, b_1 = \ldots = b_c \quad (15)$$

*For Eq.(15), if there is no constraint $b_1 = b_2 = \ldots = b_c$, then considering each row of $\mathbf{G}$ is independent,*

$$\max_{g_{ij}}\sum_{j=1}^{c}g_{ij}^2 \quad s.t. g_{ij} \geq 0, \sum_j g_{ij} = \mathbf{1} \quad (16)$$

*The optimal solution for problem (16) should be realized when $\mathbf{g}_i$ has only one element equal to 1 and the rest are 0, but the position of the element "1" is uncertain, which may lead to unstable solutions or even empty classes.*

*Thus, For Eq.(15), the additional constraint $b_1 = b_2 = \ldots = b_c$ forces the allocation of the same number of rows for each column, we can conclude that the problem $(\|\mathbf{G}\|_F^2)^{\frac{p}{2}}$ only reaches a maximum when $\mathbf{G}$ is a discrete matrix. In this case, $\mathbf{G}^T\mathbf{G} \in \mathbb{R}^{c \times c}$ is a diagonal matrix whose i-th diagonal element is the number of samples in the i-th cluster.*

*Combined with Eq. (13), we have*

$$f\left(\sum_{j=1}^{c}\frac{1}{c}b_j\right) = f(\frac{1}{c}\sum_{j=1}^{c}b_j) = f(\frac{1}{c}\sum_{j=1}^{c}n_j) = f(\frac{N}{c}) \quad (17)$$

*So we know that when Eq. (11) reaches its optimal value, $b_1 = \ldots = b_c = n_1 = \ldots = n_c = \frac{N}{c}$.* $\square$

According to Theorem 1, when Eq. (11) is optimal, $\mathbf{P} = \text{diag}(\sum_{i=1}^{N}g_{i1}, \sum_{i=1}^{N}g_{i2}, \ldots, \sum_{i=1}^{N}g_{ic}) = \frac{N}{c}\mathbf{I}$, so we convert Eq. (10) to

$$\min_{\mathbf{G}\geq 0, \mathbf{G1}=\mathbf{1}} \text{tr}(\mathbf{G}^T\mathbf{E}\mathbf{G}) - \beta\|\mathbf{G}\|_{sp}^p \quad (18)$$

where $1 \leqslant p < 2$. To simplify the optimization process, we set $f(\mathbf{G}) = \|\mathbf{G}\|_{sp}^p$ and perform a first-order Taylor expansion of $f(\mathbf{G})$ at $\mathbf{G}^{(t)}$, yielding:

$$f(\mathbf{G}) = f(\mathbf{G}^{(t)}) + \langle\nabla f(\mathbf{G}^{(t)}), \mathbf{G} - \mathbf{G}^{(t)}\rangle \quad (19)$$

where $\mathbf{G}^{(t)}$ is the solution at iteration $t$, and $\nabla f(\mathbf{G}^{(t)})$ is the derivative of $\|\mathbf{G}\|_{sp}^p$.

Denote the derivative of $\|\mathbf{G}\|_{sp}^p$ by $\mathbf{Z}$, we have

$$\mathbf{Z} = \frac{\partial\|\mathbf{G}\|_{sp}^p}{\partial\mathbf{G}} = p\mathbf{U}\mathbf{\Sigma}^{-1}|\mathbf{\Sigma}|^p\mathbf{V}^\mathrm{T} \quad (20)$$

where $\mathbf{G} = \mathbf{U}\mathbf{\Sigma}\mathbf{V}^\mathrm{T}$.

Neglecting constants in Eq. (19), we can solve Eq. (18) iteratively by:

$$\min_{\mathbf{G}\cdot\mathbf{1}=\mathbf{1}, \mathbf{G}\geqslant 0} \text{tr}(\mathbf{G}^T\mathbf{E}\mathbf{G}) - \beta\text{tr}(\mathbf{Z}^T\mathbf{G}) \quad (21)$$

Let $\mathbf{G} = \begin{bmatrix}\mathbf{g}^i \\ \mathbf{G}_0\end{bmatrix}$, $\mathbf{E} = \begin{bmatrix}e_{ii} & \mathbf{e}_{i0}^T \\ \mathbf{e}_{i0} & \mathbf{E}_0\end{bmatrix}$, where $\mathbf{G}_0 \in \mathbb{R}^{(N-1)\times K}$, $\mathbf{e}_{i0} \in \mathbb{R}^{(N-1)\times 1}$, and $\mathbf{E}_0 \in \mathbb{R}^{(N-1)\times(N-1)}$. Similarly, $\mathbf{Z} = \begin{bmatrix}\mathbf{z}^i \\ \mathbf{Z}_0\end{bmatrix}$. We have:

$$\begin{aligned}
\mathbf{G}^T\mathbf{E}\mathbf{G} - \beta\mathbf{Z}^T\mathbf{G} &= \left([(\mathbf{g}^i)^T \quad (\mathbf{G}_0)^T]\begin{bmatrix}e_{ii} & \mathbf{e}_{i0}^T \\ \mathbf{e}_{i0} & \mathbf{E}_0\end{bmatrix} - \beta[(\mathbf{z}^i)^T \quad (\mathbf{Z}_0)^T]\right)\begin{bmatrix}\mathbf{g}^i \\ \mathbf{G}_0\end{bmatrix} \\
&= (\mathbf{g}^i)^T e_{ii}\mathbf{g}^i + (\mathbf{G}_0)^T\mathbf{e}_{i0}\mathbf{g}^i + (\mathbf{g}^i)^T\mathbf{e}_{i0}^T\mathbf{G}_0 \\
&\quad + (\mathbf{G}_0)^T\mathbf{E}_0\mathbf{G}_0 - \beta\left((\mathbf{z}^i)^T\mathbf{g}^i + (\mathbf{Z}_0)^T\mathbf{G}_0\right)
\end{aligned} \quad (22)$$

Removing terms unrelated to $\mathbf{g}^i$, and using the properties of the trace operation, we have:

$$\text{tr}\left(\mathbf{G}^T\mathbf{E}\mathbf{G} - \beta\mathbf{Z}^T\mathbf{G}\right) = \text{tr}\left((\mathbf{g}^i)^T e_{ii}\mathbf{g}^i + 2\mathbf{g}^i\mathbf{G}_0^T\mathbf{e}_{i0} - \beta\mathbf{g}^i(\mathbf{z}^i)^T\right) = \mathbf{g}^i(\mathbf{g}^i)^T e_{ii} + \mathbf{g}^i\mathbf{f} \quad (23)$$

where $\mathbf{f} = 2\mathbf{G}_0^T\mathbf{e}_{i0} - \beta(\mathbf{z}^i)^T$.

Thus, the problem of updating the $i$-th row of $\mathbf{G}$ becomes:

$$\min_{\mathbf{g}^i\cdot\mathbf{1}=\mathbf{1}} \mathbf{g}^i(\mathbf{g}^i)^T e_{ii} + \mathbf{g}^i\mathbf{f} \quad (24)$$

As $e_{ii} = 0$ $(i = 1, 2, \ldots, N)$, Eq. (24) reduces to:

$$\min_{\mathbf{g}^i} \mathbf{g}^i\left(2\mathbf{G}^T\mathbf{e}_i - \beta(\mathbf{z}^i)^T\right) \quad (25)$$

where $\mathbf{e}_i$ is the $i$-th column of $\mathbf{E}$, with $e_{ii} = 0$. $\mathbf{G}$ denotes the solution before $\mathbf{g}^i$ is updated. Then, the solution of $\mathbf{g}^i$ can be written as:

$$g_{ib} = \begin{cases} 1, & b = \arg\min_j \left(2\mathbf{G}^T\mathbf{e}_i - \beta(\mathbf{z}^i)^T\right)_j \\ 0, & \text{otherwise} \end{cases} \quad (26)$$

Finally, we conclude the pseudo-code on Algorithm 2.

---

**Algorithm 2:** Pseudo-Code for BCRD

---

**Require:** Data matrix $\mathbf{X} \in \mathbb{R}^{N\times d}$; Number of neighbors
    $k$; Number of clusters $c$.
**Ensure:** Cluster assignment matrix $\mathbf{G}$.
 1: **Initialize**: $\mathbf{G}$.
 2: **while** *not converge* **do**
 3:    Update projection matrix$\mathbf{M}$ by Algorithm 1;
 4:    Update $\mathbf{Z}$ by Eq. equation 20;
 5:    Update $\mathbf{G}$ by (26) row by row;
 6: **end while**
 7: **return**: The cluster assignment matrix $\mathbf{G}$.

---

### 3.2 TIME COMPLEXITY ANALYSIS:

The time complexity of solving $\mathbf{M}$ using eigen-decomposition is $\mathcal{O}(d^3)$. For $\mathbf{G}$, the complexity is primarily due to the computation of the distance matrix $\mathbf{E}$ and the optimization of $\mathbf{Z}$ and $\mathbf{G}$ row by row. The computation of $\mathbf{E}$ has a complexity of $\mathcal{O}(N^2dm)$. Optimizing $\mathbf{Z}$ requires $\mathcal{O}(Nc^2)$ according to Eq. equation 20, and solving $\mathbf{G}$ row by row has a complexity of $\mathcal{O}(N^2)$. Thus, the overall time complexity of Algorithm 2 is $\mathcal{O}(t_1d^3 + t_2N^2dm)$, where $t_1$ represents the number of iterations of Algorithm 1, and $t_2$ corresponds to the number of iterations of Algorithm 2.

## 4 EXPERIMENTS

### 4.0.1 DATASETS

We validate the clustering performance of our algorithm using seven datasets, including: (1) AR Martinez & Benavente (1998) contains 120 face classes with 3,120 images. (2) JAFFE Lyons et al. (1998) consists of 213 images displaying different facial expressions from 10 Japanese female subjects. (3) MSRC_V2 Winn & Jojic (2005) includes 7 object classes with 210 images. We selected the 576-D HOG feature as the single view dataset. (4) ORL Samaria & Harter (1994) contains 400 facial images from 40 individuals. (5) UMIST Graham & Allinson (1998) consists of 564 facial images from 20 individuals. (6) USPS Hull (1994) consists of 3,000 handwritten digit images for each of the 10 digits from 0 to 9. (7) Yaleface Belhumeur et al. (1997b) contains 165 grayscale images in GIF format from 15 individuals.

### 4.0.2 COMPARISON ALGORITHMS

We selected several popular K-means algorithms, including K-means, Ksum Pei et al. (2023), Ksum-x Pei et al. (2023), RKM Lin et al. (2019), and CDKM Nie et al. (2022). Additionally, we included some dimensionality reduction clustering algorithms, both two-step algorithms: PCA-KM Turk & Pentland (1991), LPP-KM He & Niyogi (2003), LDA-KM Ding & Li (2007), and one-step algorithms: Un-RTLDA Wang et al. (2021), Un-TRLDA Wang et al. (2021).

### 4.1 CLUSTERING PERFORMANCE

As shown in Table 1, the clustering algorithms with dimensionality reduction outperform those in the original space in terms of clustering performance. This suggests that dimensionality reduction plays a crucial role in clustering, likely by reducing noise and emphasizing key features. Among dimensionality reduction techniques, the clustering performance of one-step dimensionality reduction algorithms surpasses that of two-step dimensionality reduction clustering algorithms. The two-step algorithms extract data features through dimensionality reduction and then use them for clustering; however, these features may not be ideal, affecting the clustering outcome. The performance of LPP-KM, based on manifold learning, exceeds that of PCA-KM, which relies on global linear transformation. LPP-KM retains the local manifold structure of the data, which often represents the intrinsic geometric structure of the data space better than global linear methods like PCA. Moreover, BCRD outperform the one-step clustering algorithm based on LDA. This superior performance is attributed to BCRD's ability to preserve the manifold structure during dimensionality reduction, allowing for a more accurate representation of inherent data clustering.

Table 1: ACC, NMI, Purity on seven benchmark datasets, with the highest value represented in bold.

| | | | | | ACC | | | | | | |
|---|---|---|---|---|---|---|---|---|---|---|---|
| Datasets | K-means | Ksum | Ksum-x | RKM | CDKM | PCA-KM | LPP-KM | LDA-KM | Un-RTLDA | Un-TRLDA | BCRD |
| AR | 0.251 | 0.297 | 0.245 | 0.264 | 0.265 | 0.282 | 0.449 | 0.258 | 0.561 | 0.272 | **0.565** |
| JAFFE | 0.709 | 0.879 | 0.893 | 0.831 | 0.711 | 0.887 | 0.977 | 0.958 | 0.967 | 0.916 | **1.000** |
| MSRC | 0.605 | 0.752 | 0.685 | 0.629 | 0.666 | 0.691 | 0.724 | 0.729 | 0.671 | 0.752 | **0.886** |
| ORL | 0.520 | 0.634 | 0.588 | 0.500 | 0.551 | 0.553 | 0.550 | 0.568 | 0.663 | 0.608 | **0.895** |
| UMIST | 0.434 | 0.421 | 0.430 | 0.421 | 0.421 | 0.464 | 0.457 | 0.499 | 0.497 | 0.501 | **0.817** |
| USPS | 0.649 | 0.766 | 0.724 | 0.667 | 0.642 | 0.778 | 0.686 | 0.759 | 0.744 | 0.791 | **0.826** |
| Yaleface | 0.381 | 0.434 | 0.442 | 0.449 | 0.396 | 0.473 | 0.400 | 0.454 | 0.473 | 0.485 | **0.655** |
| | | | | | NMI | | | | | | |
| Datasets | K-means | Ksum | Ksum-x | RKM | CDKM | PCA-KM | LPP-KM | LDA-KM | Un-RTLDA | Un-TRLDA | BCRD |
| AR | 0.557 | 0.596 | 0.568 | 0.575 | 0.570 | 0.583 | 0.704 | 0.555 | 0.801 | 0.567 | **0.804** |
| JAFFE | 0.801 | 0.876 | 0.901 | 0.816 | 0.798 | 0.914 | 0.974 | 0.948 | 0.963 | 0.923 | **1.000** |
| MSRC | 0.528 | 0.611 | 0.575 | 0.561 | 0.569 | 0.603 | 0.625 | 0.584 | 0.548 | 0.638 | **0.774** |
| ORL | 0.723 | 0.794 | 0.769 | 0.714 | 0.753 | 0.733 | 0.692 | 0.746 | 0.816 | 0.772 | **0.929** |
| UMIST | 0.641 | 0.619 | 0.638 | 0.596 | 0.640 | 0.625 | 0.662 | 0.674 | 0.693 | 0.685 | **0.890** |
| USPS | 0.615 | 0.662 | 0.608 | 0.587 | 0.606 | 0.667 | 0.653 | 0.656 | 0.642 | 0.676 | **0.701** |
| Yaleface | 0.439 | 0.498 | 0.502 | 0.510 | 0.478 | 0.508 | 0.430 | 0.508 | 0.521 | 0.527 | **0.664** |
| | | | | | Purity | | | | | | |
| Datasets | K-means | Ksum | Ksum-x | RKM | CDKM | PCA-KM | LPP-KM | LDA-KM | Un-RTLDA | Un-TRLDA | BCRD |
| AR | 0.275 | 0.369 | 0.324 | 0.322 | 0.286 | 0.299 | 0.493 | 0.278 | 0.597 | 0.294 | **0.601** |
| JAFFE | 0.746 | 0.879 | 0.898 | 0.831 | 0.744 | 0.887 | 0.977 | 0.958 | 0.967 | 0.916 | **1.000** |
| MSRC | 0.628 | 0.752 | 0.691 | 0.633 | 0.680 | 0.719 | 0.724 | 0.729 | 0.691 | 0.752 | **0.886** |
| ORL | 0.571 | 0.656 | 0.606 | 0.520 | 0.609 | 0.603 | 0.588 | 0.610 | 0.698 | 0.645 | **0.896** |
| UMIST | 0.511 | 0.455 | 0.472 | 0.440 | 0.504 | 0.518 | 0.564 | 0.553 | 0.569 | 0.562 | **0.854** |
| USPS | 0.680 | 0.768 | 0.724 | 0.683 | 0.675 | 0.778 | 0.722 | 0.759 | 0.744 | 0.791 | **0.826** |
| Yaleface | 0.403 | 0.473 | 0.492 | 0.485 | 0.419 | 0.473 | 0.424 | 0.473 | 0.497 | 0.503 | **0.661** |

### 4.2 PARAMETER ANALYSIS

We conducted sensitivity analyses on key hyperparameters: embedding dimension $m$, number of neighbors $k$, Schatten-$p$ norm parameter $p$, and weights $\beta$ and $\eta$. The $p$ value ranges from 0.1 to 1.9. The parameter $\beta$ balances $\text{tr}(\mathbf{G}^\top \mathbf{E} \mathbf{G})$ and $\beta \|\mathbf{G}\|_{sp}^p$, whose magnitudes typically differ by 100–1000×, thus requiring dataset-specific tuning (e.g., $\beta = 0.3$ for MSRC, $\beta = 8000$ for UMIST). Similarly, $\eta$ balances $\text{tr}(\mathbf{M}^\top \mathbf{X}^\top \mathbf{L}_S \mathbf{X} \mathbf{M})$ and $\text{tr}(\mathbf{M}^\top \mathbf{X}^\top (\eta \widetilde{\mathbf{D}}) \mathbf{X} \mathbf{M})$, where the first term is usually an order of magnitude larger. The optimal values are $\eta = 0.1$ (MSRC) and $\eta = 0.0004$ (UMIST).

**Discussion of the value of k:** To assess the impact of $k$ and validate the effectiveness of label-guided neighborhood similarity, Figure 1 shows clustering performance as $k$ varies from 2 to 30. The experimental results reveal that for MSRC, increasing $k$ generally enhances clustering performance, with the best results observed at $k = 14$, followed by a decline. ORL performs best when $k = 5$. In contrast, the clustering performance of UMIST reaches its peak at $k = 2$, and then gradually decreases as $k$ increases. This phenomenon can be explained by the trade-off in neighborhood selection: a larger $k$ incorporates more local information, which can help capture meaningful structural relationships. However, if $k$ becomes excessively large, the expanded neighborhood range may lead to samples from different clusters being grouped together, thereby reducing clustering accuracy.

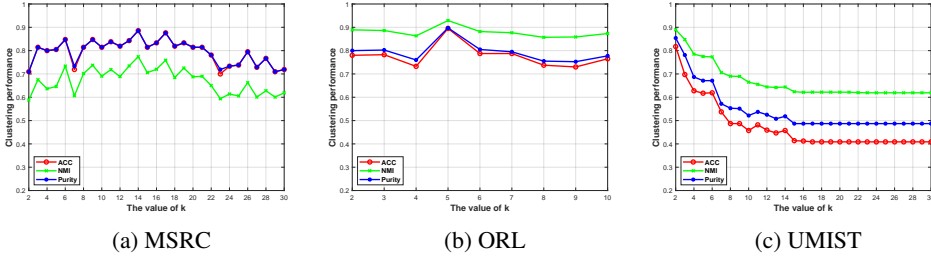

| (a) MSRC | (b) ORL | (c) UMIST |

Figure 1: The effect of $k$ on clustering performance.

**Discussion of dimensions $m$:** As shown in Figure 2, the results reveal the impact of varying dimensions on clustering performance. Our findings show that as the dimensionality increases, clustering accuracy initially improves, suggesting that appropriate dimensional representations help preserve essential structural information beneficial for clustering. However, beyond a certain point, further increasing the dimensionality leads to either performance saturation or a decline. This indicates that while a lower-dimensional representation may fail to capture sufficient distinguishing features, an overly high-dimensional space introduces redundant or noisy information, which can negatively impact clustering performance.

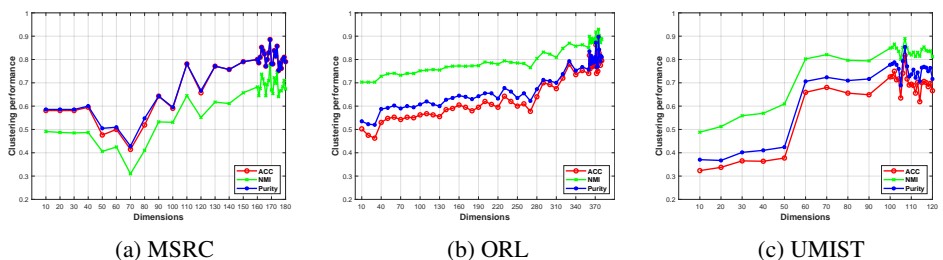

| (a) MSRC | (b) ORL | (c) UMIST |

Figure 2: The effect of $m$ on clustering performance.

### 4.3 DISCUSSION ON CLUSTERING BALANCE

To evaluate the balancing capability of our algorithm, we constructed Balanced-8 (B8) dataset. The B8 dataset consists of 8 categories, also with 100 samples per category. The experimental results are illustrated in Figure 3. By comparing the clustering distributions of the K-means and BCRD algorithms on the B8 dataset, clear performance differences can be observed. Figure 3(a) shows the clustering results of K-means, which reveal a severe class imbalance: class 5 contains only 34 samples, while class 1 dominates with 175 samples. This imbalance is mainly attributed to K-means optimizing solely the Euclidean distance without considering cluster balance, and its sensitivity to initialization often leads to uncontrolled cluster sizes. In sharp contrast, Figure 3(b) presents the clustering distribution of the BCRD algorithm, where all classes are evenly distributed with 100±3 samples each. This balance is achieved through our proposed Schatten-p norm regularization mechanism, which promotes class balance by maximizing the Schatten-p norm of the label matrix.

And we constructed an imbalanced dataset, Imbalanced-8 (ImB8), which consists of 8 categories with varying sample sizes: 50, 120, 60, 150, 100, 200, 80 and 40. Using K-means, we obtained

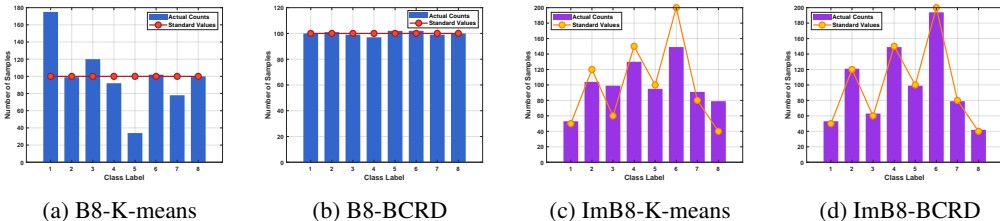

| (a) B8-K-means | (b) B8-BCRD | (c) ImB8-K-means | (d) ImB8-BCRD |

Figure 3: Comparison on B8 v.s. ImB8 Datasets.

an ACC of 0.8175. The clustering distribution is visualized in Figure 3(c). The resulting cluster sizes were 53, 104, 99, 130, 95, 149, 91 and 79, showing significant deviation from the original class distribution. In contrast, our BCRD algorithm achieved a significantly higher ACC of 0.9825. The clustering distribution is shown in Figure 3(d), where the cluster sizes were 53, 121, 59, 149, 99, 198, 79, and 42. These results show that in imbalanced datasets, our method maintains the clustering relationships of closely spaced samples, naturally preserving the original imbalanced distribution rather than forcing balance. Overall, our method achieves reasonable clustering results under different data conditions, achieving cluster balance on balanced datasets while preserving the original distribution of data on imbalanced datasets.

### 4.4 VERIFICATION OF CONVERGENCE

We analyzed the clustering performance and the objective function value as the number of iterations increases, as shown in Figure 4. From the figure, we observe that as the number of iterations increases, the clustering performance gradually improves, demonstrating the effectiveness of the iterative optimization process. Simultaneously, the objective function value exhibits a steady downward trend, indicating that the optimization procedure is minimizing the loss. After a certain number of iterations, both the objective function value and the clustering performance stabilize, providing strong empirical evidence of the algorithm's convergence. Additionally, our results highlight that the BCRD algorithm converges rapidly, requiring only a few iterations to reach a stable state.

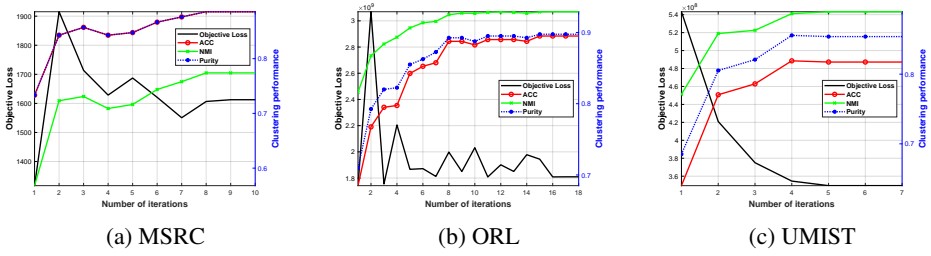

| (a) MSRC | (b) ORL | (c) UMIST |

Figure 4: Clustering performance v.s. objective loss.

## 5 CONCLUSION

This paper proposes a low-dimensional manifold balanced clustering algorithm that integrates K-means and graph embedding. A manifold structure is constructed through labels and neighborhood constraints, and the labels are optimized on this manifold to ensure the consistency between the labels and the manifold structure. Furthermore, we prove that maximizing the Schatten-p norm naturally promotes category balance in the clustering process. On balanced datasets, our method effectively promotes balanced clustering, making the sample sizes in each class more uniform. On imbalanced datasets, it naturally maintains the clustering relationship of closely related samples without forcing a balanced class distribution, ensuring that the clustering results align with the inherent structure of the data. Compared with existing clustering algorithms that operate in the original space or utilize dimensionality reduction, BCRD show significant performance improvements.

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
