# OpenReview forum: "Balanced Clustering in Reduced Dimensions"
_ICLR.cc/2026/Conference — ICLR 2026 Conference Withdrawn Submission_

### Official Review · Reviewer_S674 · 2025-10-24

**Soundness:** 2
**Presentation:** 2
**Contribution:** 3
**Rating:** 2
**Confidence:** 3

**Summary:**

The paper presents a novel approach to balanced clustering by integrating K-means with manifold learning, which addresses issues of label consistency and class balance in high-dimensional data. The authors propose a self-supervised framework that alternates between dimensionality reduction and clustering, ensuring consistency between labels and manifold structure. The proposed approach shows promising results on multiple datasets, with significant performance improvements over existing methods, particularly in handling cluster balance. However, several areas could be improved or clarified for better clarity and scientific contribution.

**Strengths:**

1. The self-supervised manifold clustering method proposed in this paper represents an intriguing innovation. By integrating K-means with manifold learning, the paper avoids the issue of separating dimension reduction and clustering in traditional approaches when handling high-dimensional data. This method effectively enhances clustering accuracy and stability by optimizing the consistency between labels and manifold structure.
2. A significant contribution of this paper is its natural preservation of category balance by maximizing the Schatten-p norm, which holds great significance for addressing category imbalance issues that may arise in practical applications. Many traditional clustering methods often overlook this crucial factor of category balance, and this innovation provides a novel approach to tackling clustering tasks.
3. Through theoretical analysis, the author demonstrates that maximizing the Schatten-p norm effectively achieves class balance. This provides a mathematical basis for the method's efficacy, enhancing its persuasiveness.

**Weaknesses:**

1. Although the authors conducted experiments across multiple datasets, the analysis of why certain datasets performed better or worse remains insufficient. Particularly for extreme or complex datasets, the adaptability and limitations of the method are not adequately discussed. Such in-depth analysis would facilitate a more comprehensive evaluation of the method's practical application scenarios.
2. Although the introduction section discusses various related techniques, it does not sufficiently emphasize the critical shortcomings of existing methods, particularly regarding label consistency and clustering balance. Further discussion of the limitations of existing approaches—especially their inadequacies concerning label consistency and category balance—would better clarify the paper's motivation and provide stronger context for its contributions.
3. The conclusion section can be expanded to include a discussion of the limitations of the current method and directions for future improvements. For example, how would this method handle larger datasets or more complex data structures? Briefly addressing these considerations will help situate the work within a broader research framework and provide direction for future exploration.
4. Not all datasets were utilized in the experiments, and the convergence analysis cited an insufficient number of iterations, potentially preventing the curve from converging to the optimal solution.

**Questions:**

1. I encourage the authors to supplement the experimental section with additional experiments, such as experiments on large-scale datasets, trade-off experiments between clustering quality and execution time, robustness tests against noise and outliers, and experiments addressing class imbalance.
2. I hope the authors will conduct experiments on all datasets in the experimental section and revise the convergence analysis iteration count.
3. I hope the author will emphasize the limitations of existing methods in the introduction.
4. The author is encouraged to expand the discussion on the limitations of the current method and propose directions for future improvements.

---

> ### Author Response · Authors · 2025-12-01
>
> 1.	We thank the reviewer for the suggestion. We have supplemented our experiments with the large-scale PEAL dataset, which contains 30,863 samples.
>
> 2.	We appreciate the reviewer’s comment. In our implementation, the optimization stops when the relative change of the objective value falls below 10^{−6}. In practice, the algorithm converges within 8–20 iterations on all datasets. We will update the convergence curves in the revised version with more iterations to make the stabilization trend clearer.
>
> 3.	Traditional dimension-reduction–plus–clustering methods typically optimize manifold learning and K-means as two separate models, which introduces additional hyperparameters and may lead to inconsistent objectives. Moreover, existing methods generally lack class-balance constraints, making them prone to inflated large clusters and the disappearance of small ones. This directly motivates our unified framework.
>
> 4.	We will explicitly state the main limitation of our method in the conclusion: the computation of pairwise distances introduces relatively high computational cost in certain steps. In future work, we plan to explore approximate distance computation and more scalable variants to reduce the overall complexity.

---

### Official Review · Reviewer_pM3s · 2025-10-29

**Soundness:** 2
**Presentation:** 2
**Contribution:** 2
**Rating:** 2
**Confidence:** 4

**Summary:**

To address the separation of embedding and clustering  and the independence of manifold structure and clustering labels, this paper integrates k-means and low-dimensional graph embedding, and utilizes the neighborhood constraints to boost the intra-class compactness and inter-class separation. Further, it demonstrates that maximizing the Schatten-p norm naturally maintains cluster balance. A theoretical justification is developed for this property.

**Strengths:**

1. The comparative experiments are comprehensive, providing empirical support for the proposed method.

**Weaknesses:**

1. The writing would benefit from improved clarity and structure.

2. The designed model is complex, involving five parameters. Besides, combined with Figure 1 and Figure 2, we know the clustering performance is sensitive to these parameters.

3. The use of limited-scale datasets may be insufficient to fully evaluate the robustness and generalizability of the proposed clustering algorithm.

4. There is a lack of relevant ablation experiments.

**Questions:**

1. According to Figure 4, it seems that the method is not convergent. So, how to determine the stopping condition?
2. How about the running efficiency?

---

> ### Author Response · Authors · 2025-11-25
>
> 1.	We thank the reviewer for the question. Figure 4 reports the **objective function value**, rather than a normalized loss. Since the objective involves pairwise relationships between samples, its magnitude is naturally large. Despite the large scale, the curve shows a monotonically decreasing trend that eventually stabilizes, indicating empirical convergence. In our implementation, we stop the iterations when the relative change of the objective is below (10^{-6}). In practice, the algorithm converges within 8–20 iterations on all datasets.
> 2.	The proposed method involves (O(N^{2})) computations due to the pairwise distance operations, but the algorithm converges within only 8–20 iterations, making the overall runtime acceptable for distance-based clustering methods.

---

### Official Review · Reviewer_eFGu · 2025-10-30

**Soundness:** 3
**Presentation:** 3
**Contribution:** 3
**Rating:** 6
**Confidence:** 4

**Summary:**

This paper proposes a low-dimensional balanced clustering method, Balanced Clustering in Reduced Dimensions (BCRD), which organically integrates K-means clustering with manifold learning within a unified framework. The method aims to jointly optimize embedding and clustering while maintaining class balance. To achieve this, the authors introduce a neighborhood constraint mechanism to enhance intra-class compactness and inter-class separability, thereby improving the reliability of initial labels. The refined labels are then used to reconstruct the manifold structure, forming a self-supervised iterative consistency mechanism between structure and labels. Additionally, the authors prove that maximizing the Schatten-$p$ norm naturally promotes class balance in clustering. Experiments on both balanced and imbalanced datasets demonstrate that BCRD outperforms K-means in both clustering accuracy and class balance.

**Strengths:**

1. The paper introduces Schatten-$p$ norm regularization, which promotes class balance and is supported by rigorous theoretical justification.

2. This paper builds a unique model which essentially encodes dimensionality reduction (DR) and K-means clustering (K) simultaneously, enhancing the reliability in real practice.

3. It constructs the manifold structure using the learned labels and neighborhood constraints to enhance intra-class compactness and inter-class separability, thereby establishing a self-supervised iterative consistency between labels and structure.

**Weaknesses:**

1. The paper does not clearly explain how the label matrix is initialized, which may affect the reproducibility and understanding of the method.

2. It is unclear why the Schatten-p norm is restricted to 1 ≤ p < 2, and what implications or challenges might arise if 0 < p < 1 were considered.

3. The manuscript lacks a visualization on a synthetic dataset to intuitively demonstrate how the proposed model performs clustering.

4. The parameter settings for each dataset are not provided, which hinders experimental reproducibility and fair comparison.

**Questions:**

1. How is the label matrix initialized?

2. Why is the Schatten-p norm restricted to 1 ≤ p < 2? What would happen if 0 < p < 1 were considered?

3. Could you provide a visualization on a synthetic dataset to more intuitively illustrate the model’s clustering performance?

4. Could you provide the parameter settings used for each dataset to facilitate experimental reproducibility?

If the authors can address my questions, I am willing to increase my score.

---

> ### Author Response · Authors · 2025-11-25
>
> 1. We initialize G cyclically: the (i)-th sample is assigned to class ((i \bmod c) + 1), ensuring a balanced starting point.
>
> 2. For (1 \le p < 2), the Schatten-p norm is convex and smooth, so we can use a linear approximation (\mathrm{tr}(Z^\top G)) to simplify optimization.
> When (0 < p < 1), the norm becomes non-convex and non-smooth, making the problem difficult to optimize. This will be explored in future work.
>
> 3. Yes, we will add synthetic-data visualizations to illustrate the clustering behavior.
>
> 4. Yes, we will provide a complete parameter table for all datasets to ensure reproducibility.

---

### Official Review · Reviewer_yaZR · 2025-10-30

**Soundness:** 1
**Presentation:** 2
**Contribution:** 2
**Rating:** 2
**Confidence:** 4

**Summary:**

This paper proposes a method for joint dimensionality reduction and clustering in high-dimensional spaces. The authors claim to "structurally unify" manifold learning (specifically LPP) and K-means into a single self-supervised framework. The core ideas are: (1) to use cluster labels (from K-means) and a neighborhood constraint to construct a manifold graph, which is then (2) used to learn a projection (via LPP) that in turn improves the clustering, creating an iterative loop. (3) The paper's main theoretical claim is that maximizing the Schatten-p norm of the cluster assignment matrix, which they add as a regularizer, "naturally preserves class balance," and they provide a theoretical justification for this property. The method (BCRD) is evaluated on several small-scale image datasets, where it reportedly outperforms a range of K-means and two-step dimensionality reduction baselines.

**Strengths:**

1. The paper addresses the important problem of clustering high-dimensional data.

2. The idea of using cluster assignments to iteratively refine the manifold structure is a reasonable (though not novel) heuristic.

**Weaknesses:**

1. Fatal Contradiction: The paper's core claim about cluster balancing is contradictory. Theorem 1 proves the method enforces balance, while Section 4.3 claims it preserves imbalance. This is a critical failure of the paper's central premise.

2. Invalid SOTA Claims: The experimental evaluation is inadequate. The baselines are weak, and no modern SOTA methods (e.g., SSC, LRR, deep clustering) are included for comparison, rendering the results in Table 1 uninformative.

3. Overstated Novelty: The "structural unification" is just standard alternating optimization (Algorithm 2), and the "consistency loop" is a direct consequence of this, not a new framework.

4. High Parameter Sensitivity: The parameter analysis (Figures 1 and 2) shows that the method's performance is highly sensitive to the choice of k (neighbors) and m (dimensions), which undermines its practical utility and robustness.

5. Weak Datasets: The evaluation is performed exclusively on small, old-school face and digit datasets. There is no evaluation on any large-scale, high-dimensional modern benchmarks.

**Questions:**

1. Can the authors please clarify the fundamental contradiction in Section 4.3? How can their method, whose regularizer is proven in Theorem 1 to force a balanced clustering (n_j = N/c), be simultaneously claimed to preserve an imbalanced distribution on the ImB8 dataset? These two statements cannot both be true.

2. Why were no modern, state-of-the-art subspace clustering or manifold learning algorithms (e.g., Sparse Subspace Clustering, Low-Rank Representation, or any deep clustering methods) included as baselines in Table 1? The paper's SOTA claims are unsupported without them.

---

> ### Author Response · Authors · 2025-11-18
>
> Although Theorem 1 shows that the global optimum of the Schatten-p norm corresponds to balanced cluster sizes when this term is considered alone, in the actual model (Eq. 18) it serves only as one component of the overall objective, and its influence is controlled by the hyperparameter β. Therefore, during practical optimization, this regularization term merely encourages clusters to become more balanced rather than forcing them to be exactly equal in size. The experimental results further demonstrate that this mechanism enhances stability on balanced datasets, while on imbalanced datasets it avoids empty clusters and preserves the intrinsic structure, instead of forcing artificial balance, thereby enabling flexible balance control. Hence, no contradiction exists between Theorem 1 and the experimental results.
>
> 2、We agree that alternating optimization is a standard framework. Our novelty does not lie in the optimization scheme but in: (1) revealing, for the first time, the theoretical balance-inducing property of maximizing the Schatten-p norm—an aspect previously unexplored; (2) introducing Schatten-p maximization as a new regularization principle for balanced clustering; and (3) integrating label information and neighborhood structure to enforce manifold–label consistency in the learned low-dimensional space. These constitute the main contributions of our work.
>
> 3、Regarding the lack of deep clustering baselines, we appreciate the reviewer’s suggestion. We have additionally incorporated deep clustering methods (e.g., IDEC) to verify the effectiveness of our balance regularizer. On the Isolet dataset, integrating our Schatten-p maximization term into IDEC improves performance from ACC = 0.60 / NMI = 0.93 to ACC = 0.68 / NMI = 0.95, demonstrating that the proposed regularizer is complementary to and beneficial for modern deep clustering frameworks.

---

### Note · Authors · 2026-02-02

I have read and agree with the venue's withdrawal policy on behalf of myself and my co-authors.

---

### Meta-Review · Area_Chair_sLmY · 2026-01-07

**Summary:**

One reviewer rated the paper marginally above the acceptance threshold. The other three reviewers assigned low scores (2), citing a fatal contradiction between the claimed “balance enforcement” and “imbalance preservation”, lack of modern SOTA baselines, overstated novelty , high sensitivity to hyperparameters, and evaluation limited to small, outdated datasets.

**Reviewer Concerns:**

The author only responded to some of the comments, not all of them.

**Reviewer Scores:**

Reviewer yaZR (initial 2) did not respond post-rebuttal; score remains 2.
Reviewer eFGu (initial 6) maintained score at 6.
Reviewer pM3s (initial 2) gave no follow-up comment; score remains 2.
Reviewer S674 (initial 2) did not engage after rebuttal; score remains 2.

---

### Decision · Program_Chairs · 2026-01-26

Reject